# Methadone or Butorphanol as Pre-Anaesthetic Agents Combined with Romifidine in Horses Undergoing Elective Surgery: Qualitative Assessment of Sedation and Induction

**DOI:** 10.3390/ani11092572

**Published:** 2021-08-31

**Authors:** Sara Nannarone, Giacomo Giannettoni, Chiara Laurenza, Andrea Giontella, Giulia Moretti

**Affiliations:** 1Department of Veterinary Medicine, Perugia University, Via San Costanzo 4, 06126 Perugia, Italy; andrea.giontella@unipg.it (A.G.); giuliamoretti89@gmail.com (G.M.); 2ADVETIA Centre Hospitalier Vétérinaire, Vélizy-Villacoublay, Île-de-France, 78140 Paris, France; giacomogiannettoni@gmail.com; 3Section of Anesthésiologie, Université de Lyon, VetAgro Sup, Marcy l’Etoile, 69280 Lyon, France; laurenza.chiara@gmail.com

**Keywords:** butorphanol, horse, induction quality, methadone, pre-anaesthesia, romifidine, sedation quality

## Abstract

**Simple Summary:**

When considering sedation or general anaesthesia in horses, a multimodal strategy is commonly preferred over a single drug. This includes the association of alpha-2 adrenoceptor agonists, phenothiazines or opioids, to improve the overall sedative and analgesic effects accordingly. However, the use of opioids alone is limited in horses due to the risk of sympathetic stimulation, central nervous system stimulation, excitement and head jerking. In some countries, butorphanol is currently the only licensed and most used opioid in equine medicine. We aimed to evaluate the pre-anaesthetic association of romifidine with either butorphanol or methadone. The two combinations were administered before induction of general anaesthesia, which included diazepam and ketamine. Evaluations involved the degree of sedation and ataxia, effects on physiological parameters, such as heart and respiratory rates, rectal temperature, quality of induction, ease of intubation and the need for top-up agents before transition to the operating theatre and the institution of a maintenance regimen.

**Abstract:**

While butorphanol is the most commonly used opioid in horses, methadone is not licensed in most countries. Our aim was to compare the effects of both drugs, combined with romifidine, regarding the quality of sedation and induction in horses undergoing elective surgery. Results indicate the suitability of both methadone and butorphanol in this patient population. Animals were scored 10 min after intravenous injection of sedatives. Despite lower overall sedation (OS) score in horses receiving methadone (*p* = 0.002), the quality and time of induction and intubation remained unchanged. None of the horses had the lowest OS score (no sedation), nor the highest score for ataxia (horse falling). Methadone induced a tendency for minor noise reaction yet minor head lowering scores, the latter being probably the most influencing parameter when scoring OS. Measured physiological parameters decreased in both groups, with greater bradycardia recorded after methadone (*p* = 0.017), including a higher incidence of atrioventricular blocks that resolved during general anaesthesia. The quality of induction was good–excellent in most of the animals. While comparisons between the degree of antinociception were beyond the scope of this study, analgesic potency might influence the choice when considering opioids as pre-anaesthetic drugs in combination with romifidine before surgery in equines.

## 1. Introduction

Alpha-2 agonists are synthetic molecules that provide reliable sedation, potent analgesia, myorelaxation and anxiolysis in horses. Their effects result from interaction with several subtypes of α1 and α2-adrenoreceptors located in the central and peripheral nervous system [1]. In equine practice, the use of alpha-2 agonists has gained extreme importance during minor clinical or diagnostic procedures because the sedative and analgesic effects decrease the overall animal responses, hence improving operators’ safety during manipulation.

The anaesthetic-sparing effects of alpha-2 agonists result from synergistic effects with other drugs, such as phenothiazines and opioids. For this reason, these drugs are frequently combined as part of a balanced anaesthetic approach to improve the quality of induction, anaesthesia maintenance and recovery from anaesthesia and to minimize the unwanted side effects of each single drug. Particularly common adverse effects seen after intravenous (IV) use of alpha-2 agonists in horses include ataxia, hemodynamic changes—that is, bradyarrhythmias—increased systemic vascular resistance and decreased cardiac output, and a reduction in respiratory rate, arterial oxygen tension and gastrointestinal motility [2].

The combination of alpha-2 agonists with opioids induces more profound and more reliable sedation than either drug alone [3]. This is a benefit during standing procedures in which ataxia is to be avoided and antinociception is desirable [4]. While opioids provide analgesia from stimulation of µ- and ĸ-opioid receptors in the brain and the dorsal horn of the spinal cord [5], horses possess a high density of µ receptors and a small population of δ-receptors in the cerebellum and cerebral cortex, which may account for their relative sensitivity to opioids’ action [6]. Given the relevant species differences in the pharmacological response, their use in horses has been limited due to the described undesirable side-effects, which are mainly physiological, such as depression of gastrointestinal motility [7,8], and especially behavioural effects [9,10,11], rather than respiratory and cardiovascular depression [12]. Central excitation and dysphoria are reported in horses, cats, pigs and ruminants after opioids administration [13]. Excitation is generally observed when opioids are given alone and in animals without pain or when high dosages are used [14]. The underlying mechanism for excitation seems secondary to indirect effects from interactions with dopaminergic and noradrenergic pathways that control motor and behavioural responses in the brain. In fact, these effects seem to diminish by the concomitant administration of acepromazine or alpha-2 agonists, which depress dopaminergic and noradrenergic transmission, respectively [15,16].

Butorphanol is the only opioid licensed for horses in Italy and the UK and is likely one of the most extensively used opioids in equine medicine worldwide. It exerts an analgesic effect through the agonistic activity at ĸ-opioid receptors and competitive antagonist and agonist activity at μ- and δ-opioid receptors, respectively [12,17]. When administered alone, butorphanol did not induce sedation at doses between 0.05–0.2 mg/kg IV, but induced ataxia, excitation [3,18], visceral analgesia [19] and supraspinal effects, which delayed and reduced pain sensation [18]. On the other hand, sedation was improved after its combination with romifidine [20], which further enhanced analgesia in standing experimental horses [3]. Moreover, in ponies undergoing field castration with ketamine anaesthesia, the IV combination of butorphanol and romifidine at 50 µg/kg and 100 µg/kg, respectively, resulted in superior sedation compared to the combination with morphine 0.1 mg/kg IV. Nevertheless, the authors in the mentioned study did not investigate whether the superior conditions were due to better sedation or greater analgesia. Indeed, morphine was found to be a suitable alternative to butorphanol in part due to its lower cost [21]. 

Methadone is a synthetic full µ-agonist with analgesic potency and pharmacokinetics similar to those of morphine. In addition, it has antagonistic activity on N-methyl-D-aspartate (NMDA) receptors as well as δ-opioid receptor activity [22] and inhibitory effects on the reuptake of noradrenaline and serotonin in the CNS [12,23]. Methadone administration at 0.2 mg/kg IV has been reported to enhance antinociception with minimal sedative effects lasting 30 min when combined with 10 µg/kg detomidine [24] and 0.05 mg/kg acepromazine [25] in experimental horses; indeed, it provided sedation and analgesia when co-administered at constant rate infusion with detomidine for standing dental procedures [26]. 

There are marketing authorizations in multiple European countries for veterinary use of methadone in dogs and cats, while it is still not commonly licensed for use in horses. Opioids that are not licensed for the use in equines in Italy include fentanyl, buprenorphine and methadone. However, these are licensed for use in dogs and cats and could be administered in equines, applying the principle of derogation of drugs through cascade (DL n 193/2006). 

To conclude, the combination of opioids and sedatives as part of a balanced anaesthetic approach has multiple positive synergistic effects in equines. Therefore, the objective of this study was to compare the effects of romifidine combined with either intravenous butorphanol or methadone on the quality of sedation and the induction of anaesthesia, as well as physiological and clinical parameters in horses undergoing elective surgery. We hypothesized that no differences would be detected between groups.

## 2. Materials and Methods

### 2.1. Animals and Scoring Procedures

The study was designed as a prospective randomized, observer blinded clinical study which was approved by the Bioethical Committee of the University of Perugia (protocol n. 2016-10). Owners signed an informed consent for each procedure.

Sixty healthy horses, classified as ASA physical status I or II, older than one year, admitted to the Veterinary Teaching Hospital of Perugia University and scheduled for elective surgery under general anaesthesia were enrolled. Animals were fasted overnight while free access to water was allowed. 

The morning of surgery, each horse received a pre-anaesthetic clinical evaluation, which enabled the recording of baseline parameters (T0) including heart rate (HR, beats/min), respiratory rate (RR, breaths/min), rectal temperature (T, C°) and the head height above the ground (HHAG), that is, the distance from the lower lip to the ground measured with a folding wooden ruler. Thereafter, animals were prepared for surgery. After clipping and aseptic preparation of the skin, a 14 SWG intravenous catheter was placed into the left jugular vein and antibiotic and anti-inflammatory treatments were administered. After shoe removal and feet and mouth cleaning, animals were brought into the padded induction box. The horses were randomly allocated (using a series of preprepared closed envelopes) to receive an IV injection of either 0.02 mg/kg butorphanol (Dolorex, MSD Animal Health S.r.l.) (group BUT) or 0.1 mg/kg methadone (Semfortan, Eurovet Animal Health B.V.) (group MET). Premedication included 0.05 mg/kg romifidine (Sedivet, Boehringer Ingelheim) administered IV over 30 s followed, after 30 s, by the selected opioid, which was diluted to a final volume of 10 mL with 0.9% sodium chloride to ensure observer blindness to treatment. The opioid was injected IV over 1 min, and thereafter animals were left undisturbed in the darkened induction box. 

Ten minutes later (T10), a new evaluation of HR, RR, T and HHAG was made right after the assessment of sedation. Quality of sedation was scored by the blinded observer using a simple descriptive scale (SDS, Table 1) modified from Nannarone et al. [27], which included overall sedation (OS), the response to noise (RN) and the degree of ataxia (AT). The assessment was always made in the same order and included first the evaluation of the OS, which was scored from 0 to 3 (0 no sedation, 3 marked sedation) as the first impression at door opening after about one-minute acclimatation of the horse to the observer, second the RN (0 no response, 3 brisk response), after hand clapping beside the horse, then HHAG, HR, RR, and T were recorded, and finally, the AT (0 no ataxia, 4 falling down) was scored while pushing and supporting the horse against the box wall during induction of anaesthesia (Table 1). 

The occurrence of other signs such as ‘head jerking’, ‘instability’, ‘tremors’, ‘penile prolapse’ and ‘sweating’ was recorded if present or not. After the assessment, induction of anaesthesia was achieved with IV diazepam (0.04 mg/kg) and ketamine (2.2 mg/kg) administered one after the other, while horses were hand-supported against the wall by three assistants: one holding and raising the head, one pushing the shoulder and one the hip. The time for effective induction, defined as the time in seconds from the end of ketamine administration and the achievement of lateral recumbency, was recorded for each animal. Quality of induction was scored from 0 to 4 (excellent to poor) according to Kerr et al. [28] (Table 2). 

After adequate neck extension and mouth gag application, the time in seconds and the number of attempts for successful endotracheal intubation were recorded. The eventual requirement of IV top up agents, such as thiopental (0.3–0.5 mg/kg) or ketamine (0.2 mg/kg), chosen at the discretion of the anaesthetist to succeed intubation (i.e., due to persisting swallowing, inadequate myorelaxation) and/or for transition to the operating theatre and starting maintenance of general anaesthesia, was recorded as well for each horse. No attempt was made to calculate the mean amount of each top-up per group as we only recorded their eventual requirement irrespective of the total dose. Once induced, horses were hoisted to a surgery table, which was then moved to the operating theatre, located about 20 m from the induction area.

### 2.2. Statistical Analysis

The sample size was estimated based on the results of a pilot study on 14 horses in each group and according to similar studies [21,25,26]. The expected mean differences between the overall score assigned in the two groups were assessed. A test power of 80% and a significance level of 5% were used. Given the possibility of potential dropouts, faced with different breeds and temperament interactions, a number equal to twice the calculated number was enrolled.

The Shapiro–Wilk’s normality test was performed to determine if the available data was well-modelled by a normal distribution and the homogeneity of variance was tested with the Fisher F test. A paired sample *t*-test was used to investigate the difference between the observations at T0 and T10. Afterwards, a two-way ANOVA with time and group as fixed effects was performed to analyse the parametric data, and Pearson’s Chi-squared test to compare categorical data of sedation and induction scores.

The *p*-values ≤ 0.05 were considered significant, but trends (*p* ≤ 0.1) were also presented and discussed. All statistical analyses were performed using R software version 4.0.5 (R Development Core Team, 2021) [29].

Data are presented as mean ± standard deviation (SD) for continuous variables, and number and percentage for categorical variables.

## 3. Results

The two groups were well matched and individual variation should not have confounded the data. A total of 60 horses were enrolled (*n* = 29 group BUT and *n* = 31 group MET). Mean age (7 ± 5 and 7 ± 4 years in BUT and MET, respectively) and weight (471 ± 108 and 482 ± 82 kg, in BUT and MET, respectively) of the animals did not differ significantly between groups. Prevalence of males compared to females was greater in both groups as the ratio intact males/geldings/females was 18/4/7 and 16/5/10 in BUT and MET, respectively.

Breeds in the study included Italian Warmblood (*n* = 15), Arab (*n* = 11), Thoroughbred (*n* = 7), Quarter Horse (*n* = 5), Frison (*n* = 2), Standardbred (*n* = 5), Maremmano (*n* = 2), Hannover (*n* = 2), Pure Espanyol (*n* = 2), Zangersheide (*n* = 2), Pony (*n* = 2), Haflinger (*n* = 1), Holsteiner (*n* = 1), KWPN (*n* = 1), Belgian Warmblood (*n* = 1) and Polish Warmblood (*n* = 1).

Orthopaedic surgeries were performed in 14 and 15 horses in group BUT and MET, respectively, while soft tissue surgeries were performed in 15 and 16 horses in group BUT and MET, respectively. 

The study was completed without any complications; none of the horse received the lowest score (0) for overall sedation, which indicated lack of sedation, nor the highest score (4) for ataxia, which indicated severe ataxia and falling of the horse (Table 3). 

Overall sedation was generally defined as moderate or marked in group BUT, and it was scored as mild more frequently in group MET than in group BUT (*p* = 0.002). A tendency toward significance (*p* = 0.07) was reported when scoring responses to noise as more than 50% of horses in group MET had a minimal reaction. No differences were found for ataxia (*p* = 0.4).

The HHAG was similar in both groups with a mean ± SD of 56 ± 20 and 50 ± 22 % reduction from baseline in groups BUT and MET, respectively.

The occurrence of side effects, such as ‘head jerking’, ‘tremors’, ‘instability’, ‘penile prolapse’ and ‘sweating,’ did not differ significantly among the groups (Figure 1). 

Physiological parameters were significantly different in all animals at T10 (Table 4) and within time between the two groups (Table 5).

Heart rate significantly decreased over time in both groups (*p* = 8.96 × 10^−8^). Moreover, HR significantly decreased in group MET compared to group BUT at T10 (*p* = 0.017); however, 21% and 13% of horses in group BUT and MET, respectively, showed second-degree atrioventricular (AV) blocks at T0, which further increased at T10 to 28% and 29% in group BUT and MET, respectively (*p* = 0.9) (Table 5). 

Respiratory rate did not show differences among groups; however, it significantly decreased over time (*p* = 1.88 × 10^−7^) and from baseline values in both groups as overall values (*p* < 0.0001) (Table 4). 

Rectal temperature was significantly higher at T10 in group BUT compared to group MET (*p* = 0.03) (Table 5).

Time for induction and for intubation were similar in both groups (61 ± 11 and 26 ± 36 s, 65 ± 37 and 20 ± 19 s in BUT and MET, respectively); likewise, attempts for intubation did not differ significantly, resulting in 2.9 ± 2.5 and 2.4 ± 1.3 attempts in group BUT and MET, respectively. Thiopental and/or ketamine top-ups for intubation and/or transition to the operating theatre did not differ among groups and was required in 38% and 35% of horses in group BUT and MET, respectively.

The quality of induction did not show significant differences; however, one horse in group MET was scored as fair (Figure 2).

## 4. Discussion

Balanced anaesthetic protocols in equines commonly include the use of opioids in combination with sedative drugs to enhance sedation, analgesia, prolong the overall duration of action while minimizing the potential for adverse effects of individual anaesthesia drugs [30]. In the present study, both butorphanol (0.02 mg/kg) and methadone (0.1 mg/kg) were satisfactory pre-anaesthetic medication when combined with romifidine (0.05 mg/kg) before induction of general anaesthesia in horses undergoing elective surgery. Therefore, our results suggest methadone as a suitable alternative to butorphanol in equines. While the overall sedation score was significantly lower in group MET than BUT, the time and quality of induction and intubation was unaffected in this group, despite the overall lower sedation.

The variable HHAG has been largely used to measure the degree of sedation induced by alpha-2 agonists in horses [31,32,33,34] and several authors have defined ‘sufficient’ sedation as when the horse’s head position is equal to or lower than 50% of the awake position [31,35]. Using this definition, Gozalo-Marcilla et al. reported that adequate sedation was achieved only 15 min after detomidine alone and for 30 min when 0.2 mg/kg methadone was combined with 10 µg/kg detomidine, whereas sufficient sedation was not achieved with the other treatments that included 0.2 mg/kg methadone alone or combined with detomidine 2.5 and 5 µg/kg IV [24]. Nevertheless, the term ‘sufficient’ should be used carefully when adding an opioid to low doses of alpha-2 agonists. In fact, opioid-linked behavioral effects may overcome the sedative effects of the alpha-2 agonist, depending on the relative dose ratio of the drugs [24]. No significant difference in the HHAG variable was detected between the two groups in our study. However, 31% (9/29) and 51% (16/31) of horses in group BUT and MET, respectively, showed a decrease in HHAG lower than 50%, notwithstanding they received score 1 (mild) as lowest OS score in 11% (1/9) and 56% (9/16) in group BUT and MET, respectively. It is likely that the degree of HHAG probably contributed to the first impression for the blinded observer when scoring OS at T10. In turn, ‘head jerking’ was recorded in the same animals with low HHAG, in 3/9 (33%, BUT) and 5/16 (31%, MET). 

Results of our study reported a tendency to a lower response to noise in horses receiving methadone as 78% of them showed minimal or no reaction, compared to butorphanol, where only 45% of horses received similar scores. Given the lower dose of methadone in group MET compared to that used by Gozalo-Marcilla and co-workers [26], the behavioural effects of opioids are unlikely to have overcome the sedative effects of romifidine in our horses. Taylor and co-workers reported that 0.02 mg/kg butorphanol IV was the lowest and more consistent dose used in their multicentre study. At this dose, butorphanol determined a clear sedative effect when combined with alpha-2 agonists without evidence of significant ataxia, which indeed was recorded after 0.1 mg/kg [36]. These results are in concordance with our results, in which 66% of horses receiving butorphanol were scored stable and without ataxia, whereas only 10% tended to cross their hindlimbs. On the other hand, 77% of animals receiving methadone were scored as stable and without ataxia and only 7% tended to cross hindlimbs. In general, it has been reported that opioids may exacerbate the alpha-2 adrenoceptor agonist-induced ataxia, and care must be taken when moving a heavily sedated horse or it may fall [37]. Romifidine is known to produce the least ataxia among the currently available alpha-2 agonists in the equine species [27,38]; its use may have contributed to the good postural stability recorded in the horses of the current study.

The quality of induction was almost good or excellent in both groups, only one horse in group MET was scored as fair, and this was probably associated with excessive noise in the induction box and a concomitant disproportionate strength when pushing the horse towards the wall for induction, as this was a young yearling. This highlights the importance of providing a quiet environment during pre-anaesthesia and induction phases as individuals could be aroused by unexpected violent noises or movements.

Simple clinical methods were used to monitor vital signs in our study, with little evidence of relevant cardiovascular or respiratory changes, given that parameters remained within normal limits in both groups. While the rectal temperature was significantly higher in horses from the BUT group, the authors consider this to be of no clinical relevance as the values remained within limits considered physiologically normal.

Regarding opioid-induced respiratory depression, it has been reported that the combination of some opioids with alpha-2 agonists potentiates respiratory depression [9,11]. In our study, the respiratory rate was equally significantly reduced from baseline in both groups. A similar effect was already described after methadone when combined with acepromazine and xylazine [39] and after buprenorphine administration [40], but values were within the physiological range, possibly reflecting the degree of sedation and lack of any anxiety. 

In general, the cardiopulmonary effects of IV butorphanol are minimal in horses and are dose related. However, there is controversy given that HR was significantly higher after butorphanol was administered to pain-free horses [41,42] when compared to baseline, reaching a peak within 50 min post-injection independently of the route of administration [43]. A similar increase over time was described for RR by the same authors [43]. On the other hand, De Rossi et al. reported that HR did not decrease from baseline after butorphanol injection, while RR decreased, and blood pressure and rectal temperature were similar to baseline values [3]. Likewise, increased blood pressure and HR have been reported simply as a resultant linked to dysphoria after a high dose of butorphanol [41], indeed Pignaton et al. speculated that low doses of opioids might stimulate cardiovascular activity unrelated to behavioural changes [44]. However, the respiratory depressant effects of butorphanol appear to be inferior to that of opioid µ agonists [12]. 

When considering the cardiovascular effects of methadone in horses, only minimal effects are observed with no physiological or behavioural changes such as excitement, sedation, inhibition of gastrointestinal motility [12,24]. As a sole agent, it increased blood pressure and HR [45], while when combined with acepromazine and xylazine, a minor influence on these parameters was produced, however AV blocks were recorded up to 65 min [39]. Moreover, when methadone was administered with detomidine to standing horses, HR significantly decreased after 5 min and second-degree AV blocks were detected but normal synus rhythm was restored within 15 min [44]. Indeed, bradycardia and AV blocks are well-known effects of alpha-2 agonists administration [46], and while they are not uncommon in healthy undisturbed horses, they could probably be enhanced when a µ agonist drug is co-administered.

In the current study, horses in group MET presented significantly decreased HR 10 min after IV co-administration with romifidine. At the same time, the occurrence of second-degree AV blocks was similarly recorded in 28% and 29% of horses in group BUT and MET, respectively. When compared to baseline, the occurrence at T10 was doubled in group MET (21% and 13%, in groups BUT and MET, respectively). Nonetheless, a quick restoration similar to that described by Pignaton et al. [44] would likely have occurred as no arrhythmia was further detected during maintenance in all horses.

Detomidine combined with methadone decreased the potential for opioid excitation and resulted in increased antinociception [25]. Similarly, both butorphanol and levomethadone, combined with detomidine, significantly increased the nociceptive threshold after electrical stimulation for up to 90 min and prolonged analgesia for up to 45 min [4]. Additionally, the enhancement of sedation and potentiation of antinociception were reported when levomethadone was co-administered with romifidine continuous infusion in experimental horses [47]. 

Opioid potency refers to the analgesic effect compared to morphine. It is likely that activities on the different opioid receptors might result in different degrees of sedation accordingly. The analgesic effect from low doses butorphanol has been reported to be three times as potent as morphine, while methadone only one to 1.5 times. However, the potency cannot be compared due to the difference in the maximal analgesic effect [5]. Based on a study using a canine tail-clamping model, butorphanol appears to be less analgesic than morphine and other pure µ-agonists [48]. On the other hand, butorphanol appears to be more effective for mild to moderate visceral pain rather than for severe or somatic pain and it is commonly used for pain relief associated with colic and minor surgery in horses [49], therefore it is likely that it would be more indicated for short-lasting surgeries requiring superficial and visceral analgesia of about 30 to 90 min [19,50,51,52]. However, when butorphanol was administered during orthopaedic surgery, significantly fewer horses required further post-operative analgesia [53]. When comparing buprenorphine to butorphanol as pre-anaesthetic medication before field castration in ponies, there was a greater rescue drugs’ requirement in horses receiving butorphanol, probably as a consequence of the better intraoperative analgesia provided by buprenorphine [54]. It is likely that methadone, being a μ-opioid receptor agonist, might be more appropriate as an opioid component of pre-anaesthetic medication before a long-lasting surgical procedure. 

The present study has some limitations. We did not include a control group without exposure to opioids. There also might be a possibility for temperament influence on sedation scores; no attempt was made to score the individual temperament of animals nor environmental conditions and horses of multiple breeds were included. Whilst these may have affected the degree of sedation, the groups were equally represented. 

Moreover, the adopted scoring system was not validated; nevertheless, similar evaluations are described in the EquiSed scale including HHAG, ‘postural instability’ and ‘auditory response’ [34]. 

Furthermore, doses of opioids used in the present study were chosen among those commonly adopted at our institution and reported in other studies [11,14,15,36,39,55]; however, equipotency remains to be verified. 

Another study limitation was not evaluating the extension of analgesic effects provided by the drug combinations throughout the surgery. Nevertheless, according to the literature [19,50,51,52,55], the drug combinations in the present study produce satisfactory analgesia for elective surgical procedures, such as in the study herein.

## 5. Conclusions

Methadone appears to be a suitable opioid in our healthy horse population undergoing elective surgery at our institution to enhance alpha-2 adrenoceptor agonist sedation before induction of general anaesthesia. Besides an apparently minor overall sedation, it tends to decrease noise response and provide a good postural stability. Bradycardia and AV blocks may occur but tend to disappear within a short time without apparent clinical relevance. However, the analgesic effects of methadone and its duration remain to be investigated, which might contribute to its final choice over butorphanol, which is currently the only opioid licensed for use in the equine species in Italy.

## Figures and Tables

**Figure 1 animals-11-02572-f001:**
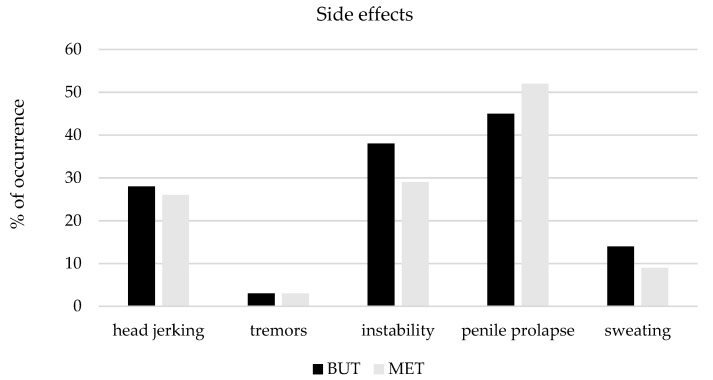
Occurrence of side effects recorded after 10 min (T10) from premedication in horses receiving romifidine and either butorphanol (group BUT, *n* = 29) or methadone (group MET, *n* = 31).

**Figure 2 animals-11-02572-f002:**
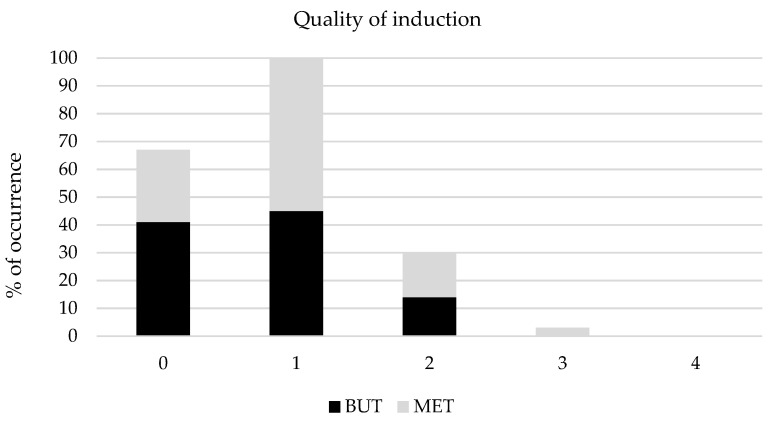
Occurrence of scores for induction quality assigned to horses receiving romifidine and either butorphanol (group BUT, *n* = 29) or methadone (group MET, *n* = 31). 0 = excellent, 1 = good, 2 = adequate, 3 = fair, 4 = poor.

**Table 1 animals-11-02572-t001:** Simple descriptive scale (SDS) scoring system for the overall sedation, ataxia, postural instability, response to noise.

Score	Variable and Description
	**Overall sedation**—first impression at door opening after one-minute acclimatation of the horse
0	No sedation, animal alert (more sedation is required)
1	Mild sedation, slightly lowered head which is raised at operator’s approach
2	Moderate sedation, lowered head, minimal response to noise
3	Marked sedation, lowered head, no response to operator’s approach and noise
	**Response to noise**—after hand clapping beside the horse
0	Brisk response, animal moves away
1	Mild response with ears and raised head
2	Minimal response with ears movements
3	No response
	**Ataxia**—evaluated by pushing the horse laterally toward the padded wall for induction
0	No ataxia
1	Stable, some swaying
2	Swaying and easily approaches the wall
3	Approaches the wall, crosses hind limbs and/or knuckles on knees
4	Falls down

**Table 2 animals-11-02572-t002:** Scoring system for the quality of induction.

Score	Description
0	Excellent—smooth induction, without muscular contraction, nor back and forward movements of the horse
1	Good—smooth induction but some back and forward movements of head and/or limbs
2	Adequate—the horse achieved recumbency without adequate limbs relaxation or after energic back and forward movements
3	Fair—notable movements or excitement and once recumbent the horse tried to stand up
4	Poor—the horse did not become recumbent

**Table 3 animals-11-02572-t003:** Parameters related to the quality of sedation assigned according to the SDS to horses receiving a premedication including romifidine and either butorphanol (group BUT *n* = 29) or methadone (group MET, *n* = 31). Values are number and percentages.

Parameter	Score	Group		*p* Value
		BUT	MET	
Overall Sedation	1 = Mild	1 (3%)	10 (32%)	*
	2 = Moderate	13 (45%)	16 (52%)	
	3 = Marked	15 (52%)	5 (16%)	
Response to noise	0 = Brisk	2 (7%)	1 (3%)	·
	1 = Mild	14 (48%)	6 (19%)	
	2 = Minimal	8 (28%)	16 (52%)	
	3 = No reaction	5 (17%)	8 (26%)	
Ataxia	0 = No	2 (7%)	6 (19%)	
	1 = Stable	17 (59%)	18 (58%)	
	2 = Swaying	7 (24%)	5 (16%)	
	3 = Crosses hindlimbs	3 (10%)	2 (7%)	

Significance codes: *: *p* ≤ 0.05; ·: *p* < 0.1.

**Table 4 animals-11-02572-t004:** Results of overall physiological parameters recorded in horses included in the study. Values were recorded at baseline (T0) and ten minutes after premedication (T10). Values are mean ± SD.

Parameter	Time		Significance (*p*-Value)
	T0	T10	
Heart rate (beats/min)	32 ± 5	26 ± 6	***
Respiratory rate (breaths/min)	16 ± 4	12 ± 3	***
Rectal temperature (°C)	37.4 ± 0.4	37.5 ± 0.5	·

Significance codes: ***: *p* ≤ 0.001; ·: *p* < 0.1.

**Table 5 animals-11-02572-t005:** Results of physiological parameters recorded in horses receiving a premedication including romifidine and either butorphanol (group BUT, *n* = 29) or methadone (group MET, *n* = 31). Values were recorded at baseline (T0) and ten minutes after premedication (T10). Values are mean ± SD.

Parameter					Significance (*p*-Value)		
	BUT_T0	MET_T0	BUT_T10	MET_T10	Time	Group	Time × Group
Heart rate (beats/min)	33 ± 6	32 ± 5	28 ± 6	24 ± 6	***	*	n.s.
Respiratory rate (breaths/min)	17 ± 3	15 ± 5	13 ± 3	12 ± 3	***	n.s	n.s.
Rectal temperature (°C)	37.5 ± 0.4	37.3 ± 0.4	37.6 ± 0.4	37.4 ± 0.5	n.s.	**	n.s.

Significance code: * *p* ≤ 0.05.; ** *p* ≤ 0.01; *** *p* ≤ 0.001; n.s. = not significant.

## Data Availability

Data presented in the study are available upon request from the corresponding author.

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
