# Peer review of "Methadone or Butorphanol as Pre-Anaesthetic Agents Combined with Romifidine in Horses Undergoing Elective Surgery: Qualitative Assessment of Sedation and Induction"

_animals, 2021, doi:10.3390/ani11092572_

Round 1

Reviewer 1 Report

Please find my comments and suggestions for the manuscript in the word document. 

Reviewer 2 Report

This study assesses the added effects of methadone and butorphanol to romifidine premedication in horses for induction of general anesthesia in horses.

The study is simple and practical and provides useful information to the reader.

Below are comments for the authors. Main comments include the statistical analysis. I have also provided comments for the different sections and topics that I believe are important for this manuscript to include.

Title:

Ok.

Simple summary:

Line 13- This sentence is confusing, as it is not common to use a single drug for general anesthesia, and the sentence implies a single drug from premedication to general anesthesia. The authors should differentiate the process of sedation, if more than one drug is elected (not necessarily preferred or better than a single drug), and then the process of general anesthesia, where multiple drugs are used for the different phases of anesthesia, and then during the maintenance period where a multimodal (balanced) drug approach can also be used.

Line 13- “When approaching sedation …” instead of current phrase.

Lines 13,14- “…a multimodal drug strategy…” instead of current phrase.

Introduction

Line 63- And the delta-receptors? Why are they not mentioned? Your reference #6, for example, alludes them in horses specifically.

This study has no hypothesis. Please state one, at least a null hypothesis.

Materials and Methods

Lines 130-131- How were these doses selected? Do the authors consider them equivalent?

Lines 135-136- Were the horses not assessed/observed for onset of sedation and other signs, during these first 10 minutes?

Lines 153,154- Please clarify and include, if the induction drugs were administered mixed in the same syringe, or if separate and the time in between them.

Line 161- What doses of thiopental or ketamine were used for the top-ups? How was it decided to pick which drug to use and when to use it, how soon after the induction? Horse with nystagmus, swallow reflex, unable to open mouth, etc. Please include.

Line 163- There is no description of what happens to the horses once induced, despite mentioning the transitioning to the operating theatre and maintenance of anesthesia. Where the horses hoisted to a surgery table or did they land on a flushed to the ground the surgery table and then elevated to ground level? This would probably influence the need for top-ups mentioned for that purpose. Please include.

Likewise, how was the top-up drug chosen for this transition period? Please include.

Statistics

You have two groups, and repeated measures. It is appropriate to compare within one group the two measurements (paired t-test); therefore, if this test was used for each group, but not to compare between groups, this analysis is acceptable.

It is not appropriate to compare two different groups for two sets of measurements, where both time and treatment are factors, to carry out within and in between groups comparisons. Therefore, your independent samples t-test is not appropriate. A two-way ANOVA should be used for this purpose.

Please include a power calculation for this study, according to your hypothesis and aims of the study.

Results:

Line 189- Do you mean “p= 0.07”, instead of “p= 0.007”. Otherwise, there is no tendency, but instead this difference is significant.

Line 201- “Physiological…” instead of “hysiological…”

Line 201- I don’t think this is the legend for Figure 1; instead, it seems to be a sentence part of the Results to allude Tables 4 and 5. Please clarify

Lines 203-205- As stated in the Statistics section above, I think the comparison between groups needs to be corrected, to see if the findings and conclusions for this table are valid.

Lines 221-223- As stated in the Materials and Methods section above, please clarify about the top-ups and their selection.

Please include if top-ups were used in the transition period.

Discussion

Line 237- add “…some physiological parameters,…” instead of current phrase.

Lines 243-246- I don’t think this was the conclusion/findings from that study. There were no significant differences in degree of sedation between 5 and 10 mcg/kg of detomidine combined with the fixed dose of methadone. In their conclusion, they even state that the lower dose of detomidine was associated with fewer adverse effects. Please correct and specify what you mean by other treatments, because the reader won’t know what those other treatments are, unless they read that reference.

Lines 247-249- Provide a reference for this statement because I am not aware of a titrating study that shows this. In general, adequate sedation with an alpha-2 that precedes the opioid is all that is recommended.

Lines 260-266- I was hoping to find in this section the reasons for selection of the doses used in this study for each opioid. As stated in my comments from Lines 130-131, please provide a section that discussed the potency and if equipotent doses were used for these opioids.

Line 306- Reference#24 reports a decrease in intestinal motility. Please include.

Please include a section on the transition period (from induction to placement on the surgery table and transfer to the theatre), as this was part of the study.

Figure 1-

As stated for Line 201, I don’t see a legend for this Figure.

Please correct the “Y: axis, so the label does not block the scale.

Figure 2-

There is no legend for this Figure.

Table 3 and Table 4

Line 195, 205- Please indicate the symbol “·” as indicated and not as a period “.”

Reviewer 3 Report

This is an interesting study reporting the quality of sedation and induction following IV administration of romifidine combined with either butorphanol or methadone in horses.

INTRODUCTION

Write a small paragraph on romifidine as it was written for butorphanol and methadone

MATERIAL AND METHODS

Specify more clearly how the study was blinded in relation to the closed envelopes

Specify the antibiotic and anti-inflammatory treatment used

Specify the horses ASA physical status

RESULTS

Insert a table of the various elective surgical procedures

Insert the comment in figure 2 (it is missing)

DISCUSSION

The sentence on line 236: “This study aimed to assess the influence on the degree of sedation and on the following induction phase of the two opioids, beside the evaluation of physiological parameters” should be moved to line 231.

Round 2

Reviewer 2 Report

The authors have addressed my previous comments adequately.